# Short wave radiative impact of liquid-liquid phase separation in "Brown Carbon" aerosols

Mehrnoush M. Fard[1], Ulrich K. Krieger[1], and Thomas Peter[1]

[1]Institute for Atmospheric and Climate Science, ETH Zurich, Zurich, Switzerland

*Correspondence to*: Ulrich Krieger (ulrich.krieger@env.ethz.ch)

**Abstract.** Atmospheric aerosol particles may undergo liquid-liquid phase separation (LLPS) when exposed to varying relative humidity. In this study, we model how the change in morphology affects the short wave radiative forcing, in particular for particles containing organic carbon as a molecular absorber, often termed "brown carbon" (BrC). Preferentially, such an absorber will redistribute to the organic phase after LLPS. We limited our investigation to particle diameters between 0.04 – 0.5 μm, atmospherically relevant organic-to-inorganic mass ratios typical for aged aerosol (1:4 < OIR < 4:1), and absorptivities ranging from zero (purely scattering) to highly absorbing brown carbon. For atmospherically relevant O:C ratios,  core-shell morphology is expected for phase-separated particles. We compute the scattering and absorption cross-sections for realistic eccentric core-shell morphologies. For the size range of interest here, we show that assuming the core-shell morphology to be concentric is sufficiently accurate and numerically much more efficient than averaging over eccentric morphologies. In the UV-region, where BrC absorbs strongest, phase-separated particles may exhibit a scattering cross-section up to 50 % larger than those of homogenously mixed particles, while their absorption cross-section is up to 20 % smaller. Integrating over the full solar spectrum, due to the strong wavelength dependence of BrC absorptivity, limits the short wave radiative impact of LLPS in the thin aerosol layer approximation. For particles with very substantial BrC absorption there will be a radiative forcing enhancement of 4 %-11.8 % depending on the Ångström exponent of BrC absorptivity for the case of small surface albedos and a decrease of up to 18 % for surfaces with high reflectivity. However, for those of moderate absorptivity, LLPS will have no significant short-wave radiative impact.

## 1 Introduction

Among many other impacts, atmospheric aerosols influence the radiation budget of the Earth directly through scattering and absorption (and less importantly emission) of incoming shortwave solar and outgoing infrared radiation. Aerosols can also affect climate indirectly through their interaction with clouds. Depending on their optical properties, size and albedo of the surface, aerosols mostly cool our planet (IPCC, 2013). However, those which are highly absorptive (e.g., soot particles) can lead to heating (e.g. Ramanathan et al., 2001, Bond et al., 2013). Anthropogenic aerosols are dominated by sulfate, organic carbon, black carbon (soot), nitrate and dust. According to the fifth Intergovernmental Panel on Climate Change report (IPCC, 2013), anthropogenic aerosols produce a net cooling effect, where radiative forcing due to aerosol–radiation

interactions is assessed to be –0.35 (–0.85 to +0.15) Wm$^{-2}$. Despite dedicated research efforts, aerosols remain one of the main sources of uncertainty for climate prediction.

Although organic aerosol particles are mainly characterized as only scattering as they are largely transparent in the visible region of the solar spectrum, a significant fraction of carbonaceous aerosols absorb solar and terrestrial radiation. Black
carbon (BC) is by far the most well known absorbing component of the atmospheric aerosols, which strongly absorbs light over a broad wavelength range from UV to IR. It has only been in recent years that a new class of organic matter was identified, which exhibit significant though weaker absorptive properties compared to BC. This absorbing fraction of organic matter is referred to as Brown Carbon (BrC) (Pöschl, 2005; Andreae and Gelencser, 2006; Ramanathan et al., 2007; Laskin et al., 2015). In contrast to BC, the absorptivity of BrC has a very strong wavelength dependence with high absorption in the
near-UV region, but absorption decreasing rapidly towards longer wavelengths (Andreae and Gelencser, 2006; Bond and Bergstrom, 2006; Ramanathan et al., 2007; Feng et al., 2013; Laskin et al., 2015). Several studies have suggested that at shorter wavelength, BrC can significantly contribute to the total aerosol absorption or even dominate it in certain geographic regions (Yang et al., 2009; Bond et al., 2011; Zhang et al., 2011; Chung et al., 2012; Feng et al., 2013; Laskin et al., 2015). Despite recent interest and extensive research regarding the impact of BrC on radiative forcing (e.g., Arnott et al., 2003;
Ramanathan et al., 2007; Alexander et al., 2008; Feng et al., 2013; Lack and Cappa, 2010; Lang-Yona et al., 2010; Lack et al., 2012; Ma and Thompson, 2012; Nakayama et al., 2010; Langridge et al., 2013; Saleh et al., 2013; Laskin et al., 2015; Tang et al., 2016), the magnitude of BrC absorption as well as its wavelength dependence is not yet well-established and its assignment to different sources and its oxidation lifetime is far from being fully characterized.

In addition to the limited knowledge associated with the optical properties of organic carbon, our current understanding with
respect to aerosol compositions, physical state, and morphology is insufficient to accurately quantify the direct radiative effect of such aerosols. In particular, particle phase and morphology need to be investigated, since they influence the scattering and absorption of radiation (e.g., Baumgardner and Clarke, 1998; Martin et al., 2004; Lewis et al., 2009; Lack and Cappa, 2010). Experiments and modelling studies have shown that as ambient relative humidity (RH) decreases, deliquesced aerosols can exist not only as a one-phase system containing organics, inorganic salts and water in a homogeneous mixture,
but often as two-phase systems, where one aqueous phase is dominated by the organic material while the other aqueous phase is predominantly inorganic, e.g. containing inorganic salts (Pankow, 2003; Marcolli and Krieger, 2006; Ciobanu et al., 2009 and 2010; Bertram et al., 2011; Krieger et al., 2012). This phenomenon is referred to as liquid-liquid phase separation (LLPS). The relative humidity (SRH) at which the transition from well mixed to liquid-liquid phase separated occurs depends on the O:C ratio of the organic as well as on the nature of the inorganic salts but typically occurs in a range between
70 % RH and 95 % RH (Song et al., 2012, You et al. 2012). In recent years, laboratory studies using model mixtures to represent tropospheric aerosols (Ciobanu et al., 2009; Bertram et al., 2011; Song et al., 2012a, Song et al., 2012b), or secondary organic aerosol (SOA) produced from smog chamber experiments (Smith et al., 2012) and filter samples collected

during field measurement campaigns (You et al., 2012) imply that liquid-liquid phase separation (LLPS) is a common feature in mixed organic/inorganic particles. When two aqueous phases coexist in a particle, they may form different morphologies, such as core-shell or partially engulfed, depending on which configuration yields the lowest total surface free energy (Kwamena et al. 2010, Qiu and Molinero, 2015). According to Song et al. (2012b) and (2013), for aged aerosols with
moderate to high oxygen-to-carbon (O:C) ratio, core-shell is the dominated morphology. For further information, see the Faraday Discussion on this very topic (Faraday Discuss. 2013, 165). Besides consequences for hygroscopicity (Hodas et al., 2015), a core-shell configuration alters the optical properties of the particles in particular for organic phases containing absorbing molecules, such as BrC since the absorbing BrC material will always reside in the organic shell.

Brown Carbon is referring to the light-absorbing fraction of the organic carbon that has a wavelength dependent imaginary
part of the refractive index, which increases towards shorter wavelengths. Emission sources of BrC are not very well characterized. The primary emissions are mainly linked to biomass burning, smoldering combustion and biogenic emissions from humic matter, plant debris and other bio-aerosols (Andreae and Gelencser, 2006; Alexander et al., 2008; Chakrabarty et al., 2010; Kirchstetter and Thatcher, 2012). Field measurements have also associated BrC to secondary organic aerosol (SOA) that form by gas to particle partitioning of semi-volatile organic compounds presenting in the biomass burning smoke
(Hecobian et al., 2010; Saleh et al., 2013). As SOA ages through oxidation processes, it may become significantly more absorbing in the near-UV region of the solar spectrum (Bones et al., 2010; Updyke et al., 2012; Laskin et al., 2015), implying that heterogeneous chemistry producing BrC in the condensed phase.

Despite the fact that the presence of LLPS has been observed and studied by a number of research groups, its impact on the radiative properties of mixed aerosol particles with molecular absorbers has so far not been quantified. In previous optical
modeling studies, mostly focusing on the optical properties of particles containing soot inclusions, typically volume mixing approximations for the optical properties were employed or the morphology were assumed to be that of a spherical symmetric, concentric core-shell. In this paper, we apply a Mie-code developed for calculating the scattering properties of a non-symmetric cluster of spheres (Mackowski, 2013, http://eng.auburn.edu/users/dmckwski/scatcodes/) to calculate the ratio in optical efficiencies between homogeneous and phase-separated particles. We vary size, absorptivity of BrC, organic to
inorganic ratio over ranges typical for aged atmospheric aerosol in the accumulation mode and show first, that the average optical efficiencies of an ensemble of phase separated particles with a random eccentric inclusion are well represented by those calculated for a simple concentric core shell particle. Second, we take advantage of this finding and calculate the radiative forcing caused by phase separated aerosol particles relative to homogenously mixed once in a thin aerosol layer approximation (Nemesure and Schwartz, 1998).

## 2 Eccentric versus concentric core shell morphology

We first want to discuss the difference in scattering and absorption accounting for differences in morphology for liquid-liquid phase separated particles. In particular, we investigate the difference between a symmetric core shell morphology compared to one with an eccentric inclusion. To represent typical aged atmospheric aerosol containing BrC we choose the

inorganic salt to be ammonium sulfate, representing the most abundant inorganic salt in continental aerosols, and light-absorbing organic carbon material representing BrC. We choose to study three organics to inorganic ratios (OIR), 1:4, 1:1, and 4:1, from inorganic rich to organic rich, which cover the typical range observed with aerosol mass spectrometry (AMS) (Zhang et al., 2007).

To account for the absorptivity of BrC, we take the imaginary part of the refractive index ($k$) for BrC spanning a wide range

from non-absorbing organic material ($k = 0$) to highly absorbing organic matter ($k = 0.168$ at 355 nm). This range is based on various studies (Kirchstetter et al., 2004; Chen and Bond, 2010; Feng et al., 2013, Wang et al., 2014, Moise et al., 2015) that measured or collected data of $k$ for different absorbing aerosol at different locations. The real part of the refractive index ($n$) for BrC at dry condition is taken as 1.65 (Hoffer et al., 2006). We use simple volume mixing to calculate the real part of BrC at 70 % RH, the size of the core relative to the shell for the different OIRs (see Fig. 1) as well as to calculate the refractive

indices for the phase-separated particles (see Tables A1 and A2 in Appendix A). (Note, we use the volume mixing approximation just to illustrate the effect of morphology in this section, for this purpose any effective medium approximation could be used.)

For about 20 years numerical calculations for scattering and absorption of a host sphere containing a non-concentrically

positioned smaller sphere have been computationally feasible. In particular, the T-matrix approach of Mackowski and Mishchenko (1996) and (2011) solves the problem of obtaining random-orientation properties of clusters of spheres in a numerically efficient manner. In our context, it has been applied in recent years for computing scattering and absorption of morphologically complex soot containing aerosol (e.g., Mishchenko et al., 2013, Cheng et al., 2014). Here, we use the Multiple Sphere T Matrix (MSTM) version 3.0 (Mackowski, 2013, http://eng.auburn.edu/users/dmckwski/scatcodes/) to

compute fixed and random oriented scattering and absorption cross sections as well as the asymmetry parameter for eccentric core shell liquid-liquid phase separated aerosol with a molecular absorber in the organic phase.

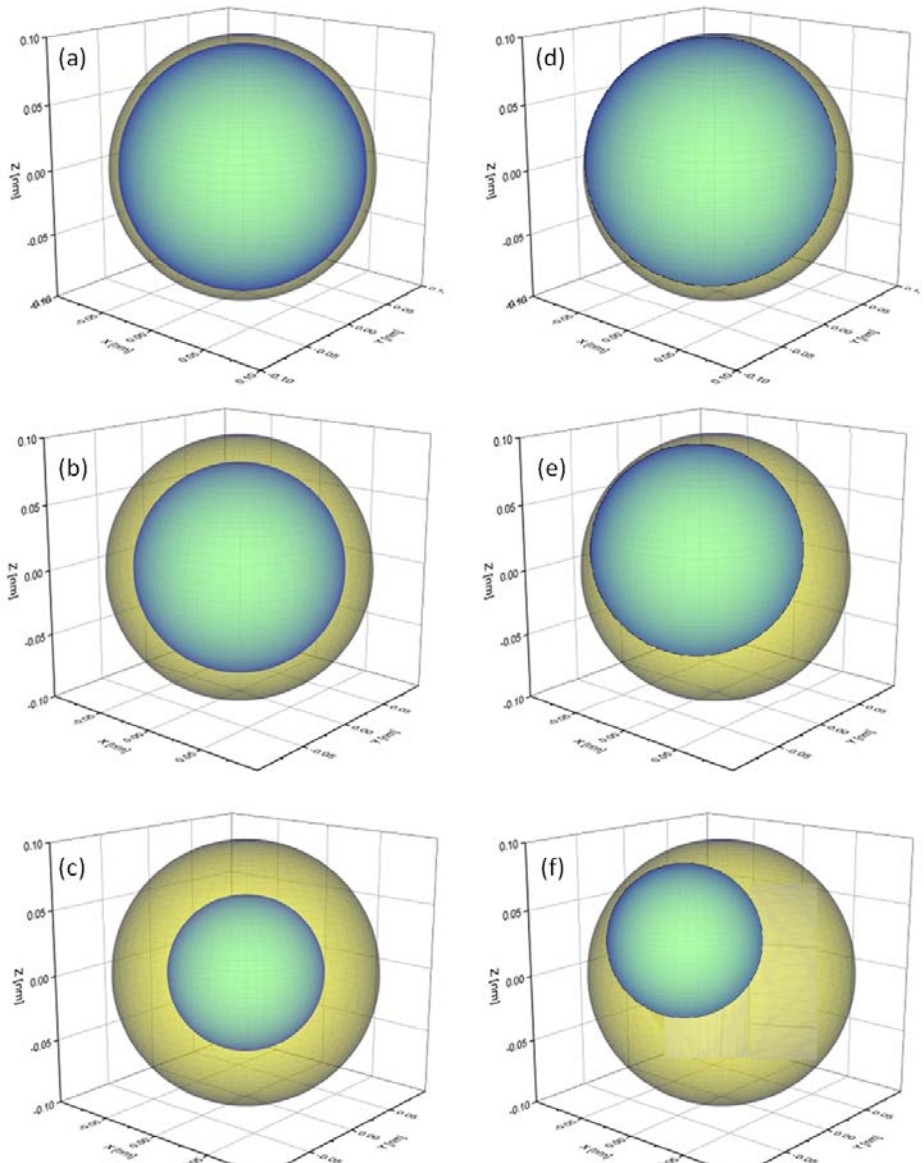

**Figure 1: Schematic of a phase separated particle with a 100 nm radius to illustrate the thickness of the organic shell at (a): OIR = 1:4, (b): OIR = 1:1, and (c) OIR = 4:1 for concentric core-shell morphologies. (d) - (f) same for a single realization of an eccentric core-shell morphology. The calculations in this work use typically 500 such realizations, randomly oriented (with different distances of the inclusion from the center of the particles and different azimuth and polar angles) .The direction of incoming light is along the *z*-axis.**

The computational costs of calculating cross sections increase substantially, when going from highly symmetric core shell morphology, to a given eccentric core position relative to the incident light, and even more when random orientational

averaging with random positioning of the core within the shell volume is required. Hence, we investigated systematically how the orientational averaged cross-sections for the eccentric morphologies differ from the cross section for the concentric morphology. We performed this comparison for particles with properties ranging within the limits of OIR, size and absorptivity discussed above.

First, we used the MSTM code with random positioning of the center of the inorganic core within the organic shell to check how many realizations of fixed positions are needed for convergence of the mean cross sections. We did two types of randomized calculations for the position of the core within the shell. (1) Random position attached to inner surface: the core remains always attached to the inside surface of the particle, hence in a spherical coordinate system the radial distance

between center of core and center of the particle remains fixed. We used a random number generator to draw random numbers for both, the polar and the azimuthal angle to place the core within the particle in the spherical coordinate system. The light is always parallel to the z-axis of a corresponding Cartesian coordinate system. (2) Random position within the volume: if the core is not attached, we also varied the distance between core center and particle center, i.e. the radial coordinate in the spherical coordinate system, by using a random number scaled such that the core access the volume within

the particle with equal probability.

Figure 2a shows the distribution of scattering efficiency ($Q_{scat}$) for a particular choice of particle parameters comparing 10000 (locations) realizations (red columns, which refer to the $Q_{scat}$ of individual particles with their core located at random

positions within the volume of the shell) with 500 realizations (green columns).
Scattering efficiency ($Q_{scat}$) is the ratio of the scattering cross-section, $\sigma_{scat}$ to the geometrical cross-section of the particle, $\sigma_{geometric}$:

$$Q_{scat} = \frac{\sigma_{scat}}{\sigma_{geometric}},$$

for spherical particles with radius $r$, $\sigma_{geometric} = \pi \, r^2$.

As illustrated in Fig. 2b the mean of the scattering efficiencies converges rapidly with the number of the realizations used to calculate this mean. The same holds true for absorption efficiencies. Conservatively, in the following we use 500 different positions of the inclusion within the particle to determine the averaged $Q_{scat}$ and $Q_{abs}$ for the particles with eccentric core-shell morphology.

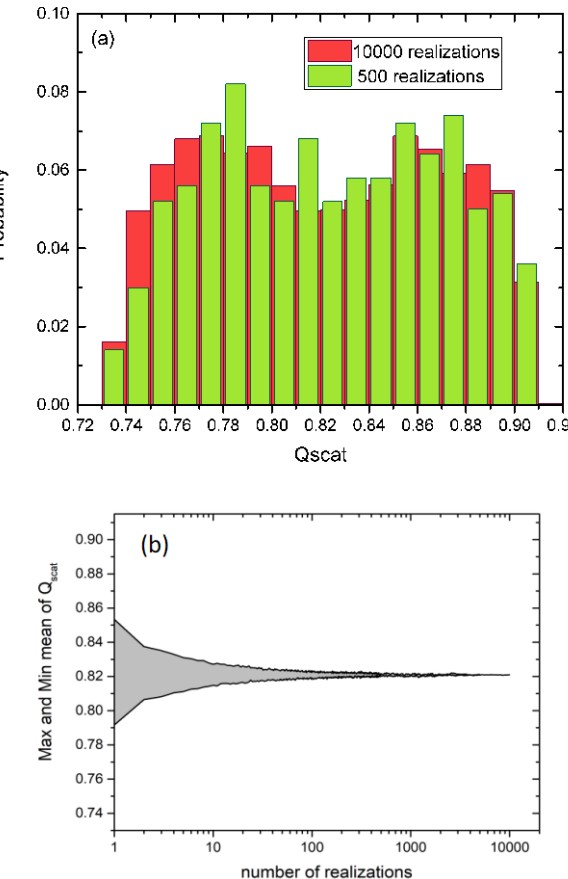

**Figure 2 (a): Probability distribution of $Q_{scat}$ for a phase separated particle with the eccentric core positioned at 500 random locations (green columns) and 10000 random locations (red column) within the volume of a 100 nm diameter particle at OIR = 1:4 n(core) = 1.429 +i·0, n(shell) = 1.571+i·0.84. (b): Maximum and minimum of the mean for $Q_{scat}$ for random realization of the position with the number of realizations used for the calculation.**

It is evident from Fig. 2a that there is a considerable range in scattering efficiency with the position of the center of the inclusion as the width of the distribution is more than 20 % of the mean of the scattering efficiency. Figure 3 shows $Q_{scat}$ versus the core center perpendicular to the direction of the incoming radiation and along the direction of the radiation. (Please, note that the incoming radiation is randomly polarized for all calculations). Obviously, the scattering efficiency changes significantly with the position of the core along the incoming light axis. It is smallest for the core position facing the incoming light and largest with the core being located at the opposing position to the incoming light with an almost linear dependence for at least this particular set of parameters. The almost linear dependence of the efficiency along the light axis and random dependence perpendicular to it suggests that a concentric core-shell calculation of the scattering efficiency may

be an excellent approximation for the mean of a mono-disperse particle ensemble with random eccentric positions. In addition it suggests that the mean of the efficiency for an inclusion randomly distributed in the volume of the particle is not very different from one where the inclusion sticks to the surface of the particle at random positions.

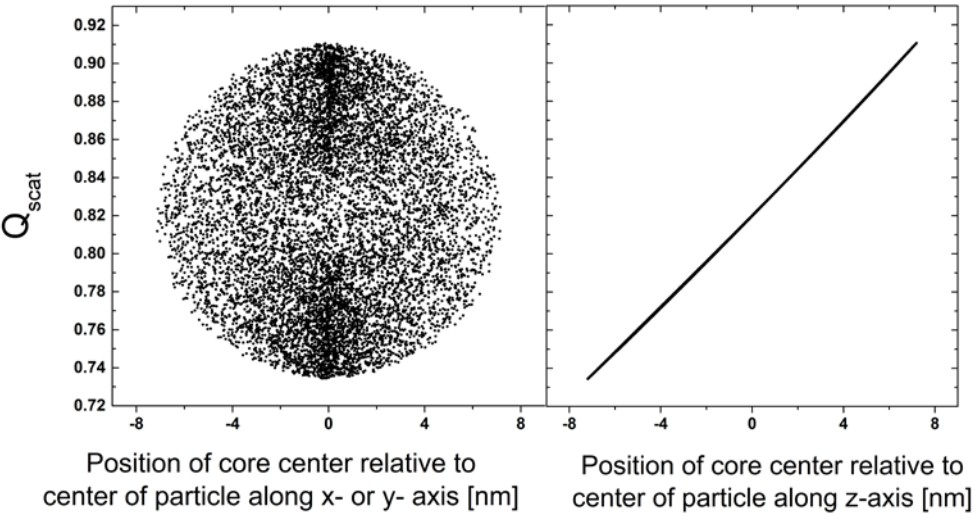

Figure 3. The change in $Q_{scat}$ with the relative position of the core ($r_{core}$ = 92.8 nm) to the particle center perpendicular to the direction of light, x- and y-axis (panel a) and along the direction of light , z-axis (panel b) for particle with OIR = 1:4, $k$ = 0.168, $r_{particle}$ = 100 nm over 10000 realizations.

For these particular particle parameter choices, this is really the case as Fig. 4 shows the mean scattering efficiency for the eccentric morphology as well as the one for a concentric core-shell with the standard deviation on an enlarged scale. The mean values for eccentric core-shell (with its center randomly placed in the volume) and eccentric core-shell (with its center position randomly placed such that it touches the surface) are very similar and only differ in the 4[th] decimal place and the value for a concentric core-shell is about 0.5 % lower. This emphasizes that the concentric core-shell model may be a good approximation for calculating the mean value for a distribution of particles with randomly located eccentric cores within either volume or surface for the OIR range and refractive indices typical of aged aerosol and particles in the accumulation size range. Let us note parenthetically, that very similar behavior is exhibit when plotting the absorption efficiency instead of the scattering efficiency.

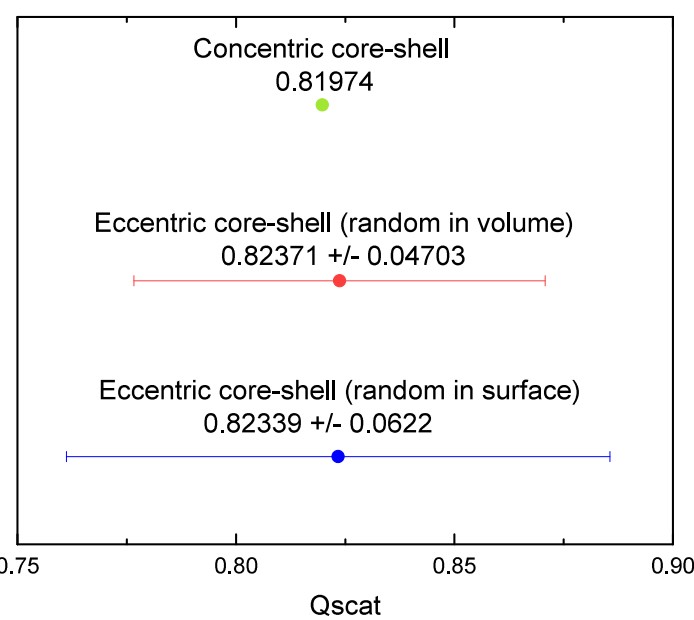

**Figure 4.:** $Q_{scat}$ **mean value and standard deviation (SD) for eccentric core-shell (over 10000 realizations) within the volume (blue),** $Q_{scat}$ **mean value and SD for eccentric core-shell (over 10000 realizations) randomly attached to the inner surface (red), and** $Q_{scat}$ **for concentric core-shell (green).**

To test this hypothesis over a wide parameter range, we compare mean scattering and absorption efficiencies for particles with eccentric core-shell morphology (with the inorganic core randomly placed at 500 different positions within the volume of the particle) with the corresponding concentric core-shell morphology.

10  As the efficiencies for scattering and absorption are strongly dependent on the size of the particle (Bohren and Huffman, 2008; Van de Hulst, 1957) for accumulation mode particles, we need to come up with a relative measure for the comparison of the two morphologies. The dependence is illustrated in Fig. 5 for a particle with $k = 0.168$, and OIR = 1:4. Here, the homogeneous particle is more efficient in absorbing the incoming light as particle size is increasing compared to the equivalent phase-separated one. In contrast, the phase-separated particle scatters light about 20 % more efficiently for

15  particles above 400-nm diameter.

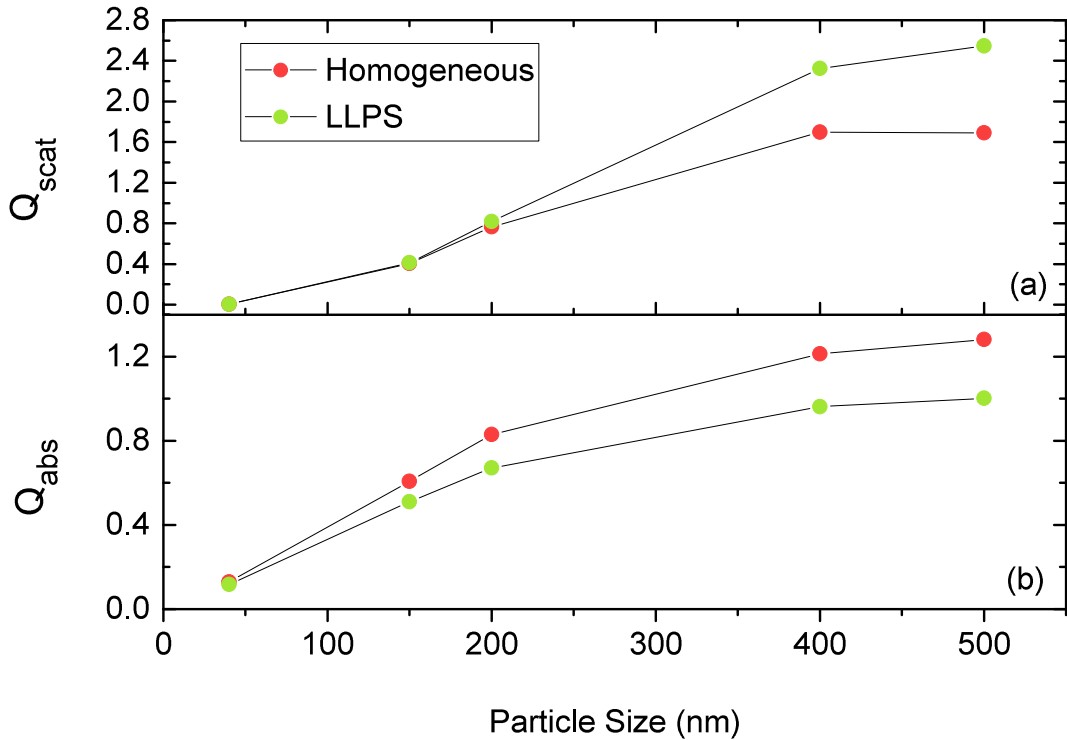

**Figure 5:** $Q_{scat}$ **(panel a), and** $Q_{abs}$ **(panel b) calculated for phase separated (LLPS) and homogeneous particles ranging from 40-500 nm at OIR = 1:4, and** $k$ **= 0.168 using the eccentric core-shell Mie-code at** $\lambda$ **= 355 nm. The red and green dots show the results for the same system at specific particle size for homogenous and phase separated (LLPS) particles respectively. The lines are only meant to guide the eye.**

Since we are interested in the impact of LLPS on aerosol scattering and absorption, we take the internally mixed, homogeneous particle as reference and plot in the following always the ratio of the LLPS morphology to the homogeneous particle for all calculated efficiencies. This way, the strong size dependence of the efficiencies seen in Fig. 5 for both morphologies cancels each other out and the emphasis is put instead on the effect of morphological change. These ratios may be understood as an empirical factor that could be used to correct calculations for homogeneous particles if those of equivalent phase separated particles are needed.

In Figure 6, we have calculated the ratio of scattering efficiencies for the LLPS morphology over the ones for the homogenous morphology for both, the eccentric case and the concentric case. As the computations for a concentric core-shell morphology are computationally very efficient, those were calculated for a small spacing in particle size whereas the calculations for the eccentric morphologies were done only for a few particles sizes. The analogues ratios for absorption efficiencies are plotted in the right column.

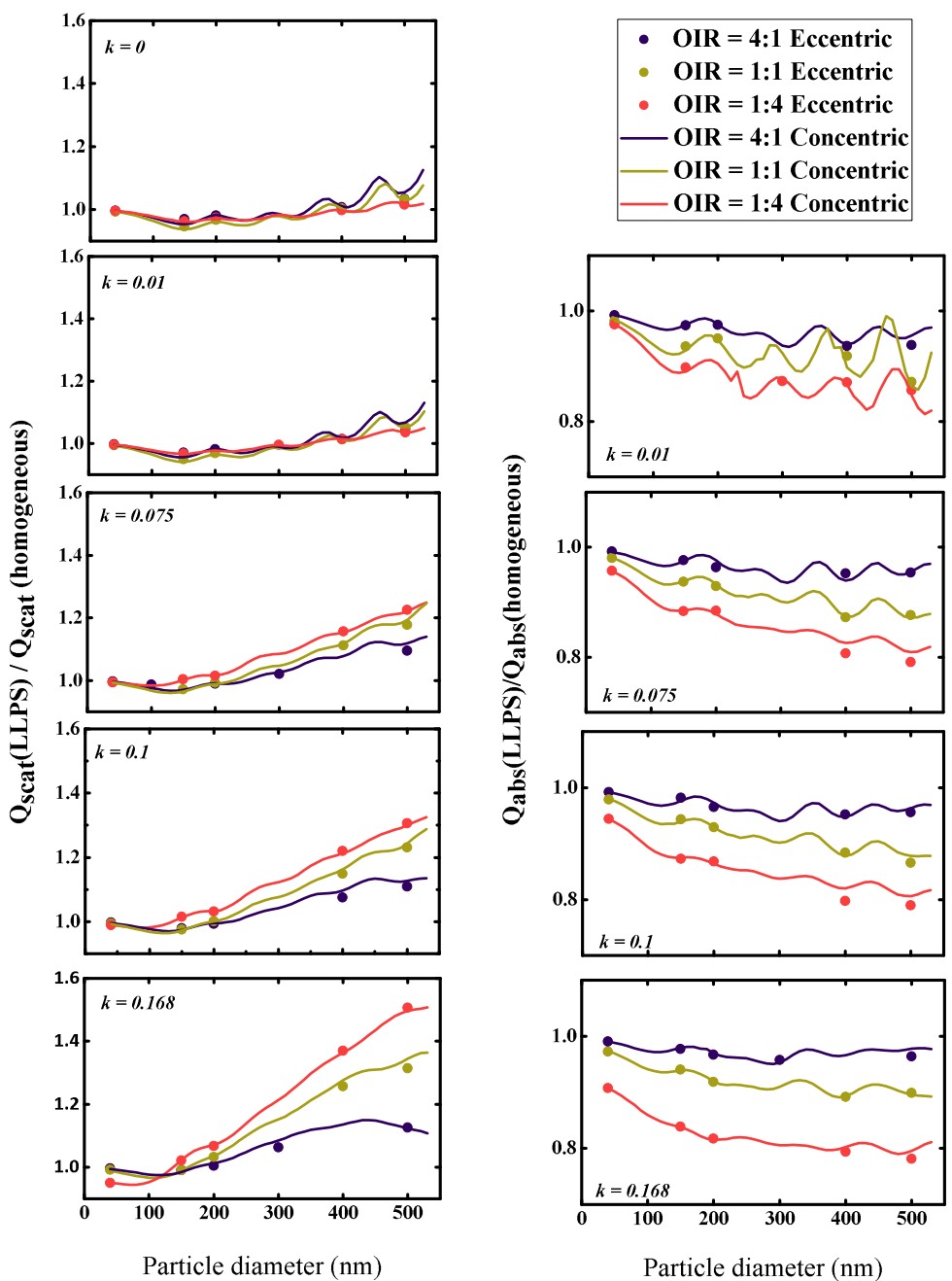

**Figure 6.** Left row: ratio of $Q_{scat}$ for LLPS morphology over homogenous morphology as function of particle diameter for OIR = 1:4, 1:1, 4:1, with increasing $k$ from top to bottom, $k$ = 0, 0.01, 0.075, 0.1, and 0.168. Right row: corresponding absorption efficiency ratios. Dots represent calculations with random orientation of an eccentric core shell for the phase-separated particles;

**lines are calculations for the corresponding symmetric core-shell morphology (we used volume mixing to calculate refractive indices, see Tables A1 and A2).**

Clearly, the effect of phase separation increases with particle size: the ratio of scattering efficiencies increases and is for all but the smallest sizes larger than one, whereas the ratio of absorption efficiencies decreases and is always smaller than one. All calculations have been performed for a wavelength of 355 nm, hence particles much smaller than this wavelength are close to Rayleigh scattering and become less sensitive to internal morphology. Hence, we expect all ratios to approach 1 for very small particle sizes. Also apparent are periodicities in both the ratio for scattering as well as the ratio of absorbing efficiencies showing a period with size of about 100 to 150 nm presumably related to half-integral ratios of size to wavelength. These dampen out with increasing absorptivity.

However, the overall effect of increasing absorptivity is to enhance the differences between phase-separated particles relative to homogeneous ones.

The most significant trend is the dependence of the efficiency ratios on the organic to inorganic ratio. For the absorption efficiency, the particles with the lowest organic volume (OIR = 1:4) show the strongest deviation for the LLPS morphology relative to the homogenous particle. As the organic fraction decreases in the particle, the effect of redistribution of the absorbing molecules into the organic phase yield a stronger contrast in the imaginary part of the refractive index between shell and core. This increase in contrast influences both absorption and scattering efficiency ratios (see Appendix A).

Most importantly, the extensive comparison between mean of eccentric core-shell realizations with concentric core shell calculations indicate that a concentric core-shell model is sufficient for estimating the ratios between scattering and absorption efficiencies for particles smaller than 500 nm in diameter and the ranges in OIR, and absorptivity under consideration here. This approximation becomes less accurate with increasing particle size but stays within 2.8 % at maximum and is better than 1 % for most of the parameter range relevant here.

## 3 Atmospheric implications

In the previous section, we showed that concentric core shell calculations are sufficient to approximate the radiative impact of LLPS for a typical atmospheric aerosol containing a molecular absorber like Brown carbon. Utilizing this insight allows us to perform integration over the UV-VIS part of the solar spectrum in a numerically efficient manner. In this section, we calculate the ratio of radiative forcing caused by a phase separated versus a homogeneously mixed aerosol in the thin aerosol layer approximation for mono-disperse aerosol.

According to Chýlek and Wong (1995) (see also Nemesure and Schwartz, 1998; compare to Charlson et al., 1991 for a purely scattering aerosol), the intrinsic properties that dictate the shortwave direct radiative forcing in the thin aerosol layer approximation for absorbing aerosol particles are their scattering and absorption cross-sections and the fraction of radiation

scattered by aerosol into the upper hemisphere, the up-scattering fraction. Here, the ratio of scattering efficiency to extinction efficiency, the single scattering albedo (SSA) $\omega$, determines the portion of total extinction due to scattering (e.g.: Moosmüller and Sorensen, 2018):

5 $$\omega = \frac{Q_{scat}}{Q_{ext}} = \frac{Q_{scat}}{Q_{scat} + Q_{abs}}$$ (1)

For the examples of Fig. 6, the ratio of the single scattering albedos of the two morphologies are shown in Fig. 7.

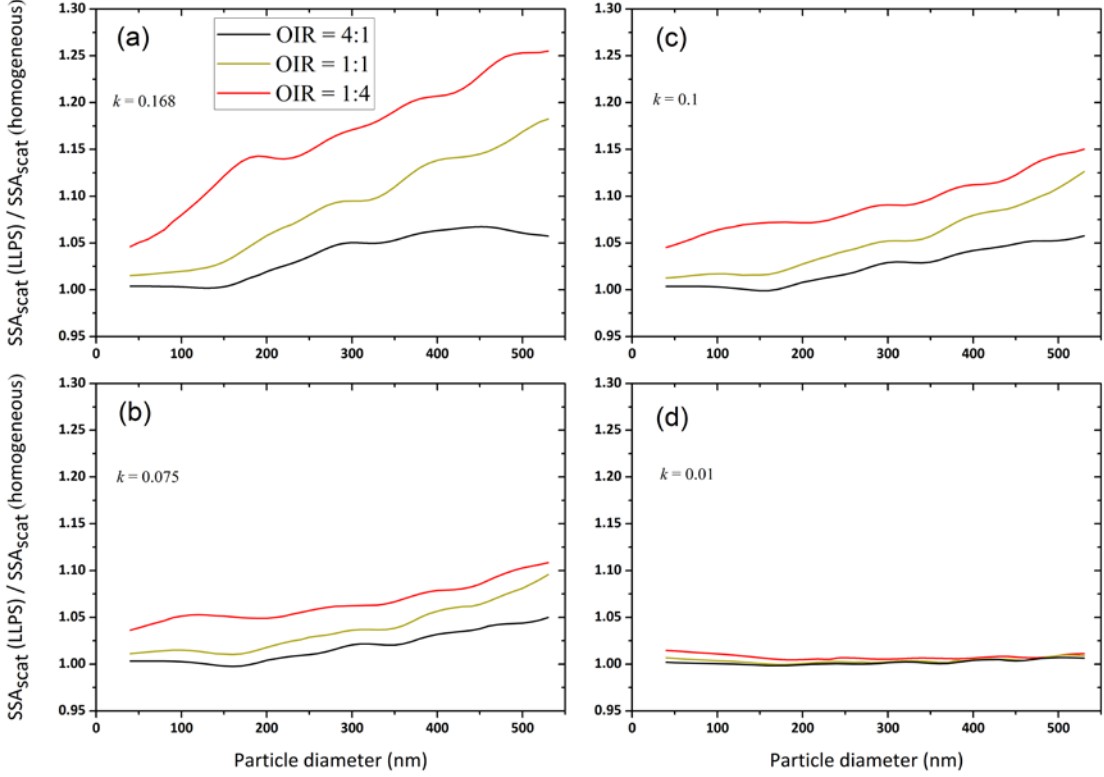

10 **Figure 7. Ratios of *SSA* for LLPS morphology over homogenous morphology as function of particle diameter for OIR = 1:4, 1:1, 4:1, with decreasing absorption from (a) to (d), *k* = 0.168, 0.1, 0.075, 0.01 (same parameters as in Fig. 6, for details, see Tables A1 and A2). Here, we show only the data calculated for concentric core shell morphologies.**

For all OIR and absorptivities, a phase-separated particle has a larger single scattering albedo compared to a corresponding homogeneous particle, up to 25% larger for the strong absorbing case and a large particle diameter. However, for weakly absorbing particles (k <= 0.01) the effect is negligible, as expected. As in Fig. 6, the strongest enhancement is observed for the OIR 1:4 case, i.e. the one with the largest redistribution of absorbing molecules upon LLPS.

Following Chýlekand Wong's (1995) line of argumentation, we calculate the direct radiative forcing, $\Delta F_R$, of an optically thin aerosol layer in a cloud free atmosphere (per unit area and unit vertical height, $\Delta z$) as:

$$\Delta F_R = -\frac{S_0}{4} \sigma \{(1-a)^2 2\beta Q_{scat} - 4aQ_{abs}\} \Delta z \tag{2}$$

With $\frac{S_0}{4}$ being the globally averaged solar flux at the top of the scattering volume, $\sigma$ the geometric cross section, $a$ being the surface albedo and $\beta$ the up-scatter fraction. The up-scatter fraction, $\beta$, is a function of particle size and accounts for the asymmetry of the scattering phase function. It has a value of 0.5 for small particles in the Rayleigh regime and decreases as the size of the particle increases. The up-scattering fraction for accumulation-mode particles ($0.1~\mu m < r < 1\mu m$) that dominates aerosols mass and light scattering properties in the atmosphere, $\beta$ may be approximated for isotropic incoming radiation by $\beta = \frac{1}{2}\left(1 - \frac{7}{8}g\right)$ (Wiscombe and Grams, 1976), with $g$ being the asymmetry parameter, i.e. the average cosine of the scattering angle ($g = \int_{4\pi} P\cos\theta\, d\Omega$, $P$ being the normalized phase function). Since we are only interested in calculating the ratio of the radiative forcing for the LLPS morphology relative to homogenous morphology, we use this approximation for the up-scatter fraction and calculate the ratio of the short wave radiative forcing for the different morphologies as:

$$Ratio\ \Delta F_R = \frac{\int_{\lambda 1}^{\lambda 2} \Delta F_R^{LLPS}(\lambda)\, d\lambda}{\int_{\lambda 1}^{\lambda 2} \Delta F_R^{Hom}(\lambda)\, d\lambda} \tag{3}$$

Let us first discuss the case for a perfectly absorbing surface, i.e. albedo $a$ equal zero. The last term in the curly bracket of Eq. (2) vanishes. The relevant factors of Eq. (2) for this albedo are shown in Fig. 8 for a particle for which we expect a significant effect of morphology based on the results presented in Fig. 6. Its OIR is equal to 1:4, it has a diameter of 200 nm, an imaginary part of the refractive index of $k = 0.168$ at 355 nm. We take the wavelength dependence of the imaginary part of the refractive index (see Appendix B) into account by using a single Ångström exponent (AAE) in the following power law relationship (Moosmüller et al., 2011):

$$k(\lambda) = k_{355}\left(\frac{\lambda}{\lambda_{355}}\right)^{-AAE} \tag{4}$$

In the example shown in Fig. 8, AAE is equal to 2 (see Fig. B3 for $k(\lambda)$ in Appendix B).

We also need to estimate the real part of the refractive index for a typical aged aerosol particle. Her, we assume it to consist
5    of aqueous ammonium sulfate and secondary organic matter. The Lorenz-Lorenz relation (Born & Wolf, 1959) is utilized to
estimate the real part of the refractive index based on parameterizations for the refractive index of ammonium sulfate and the
organic matter for dry conditions and for 70 % RH as explained in detail in Appendix B.

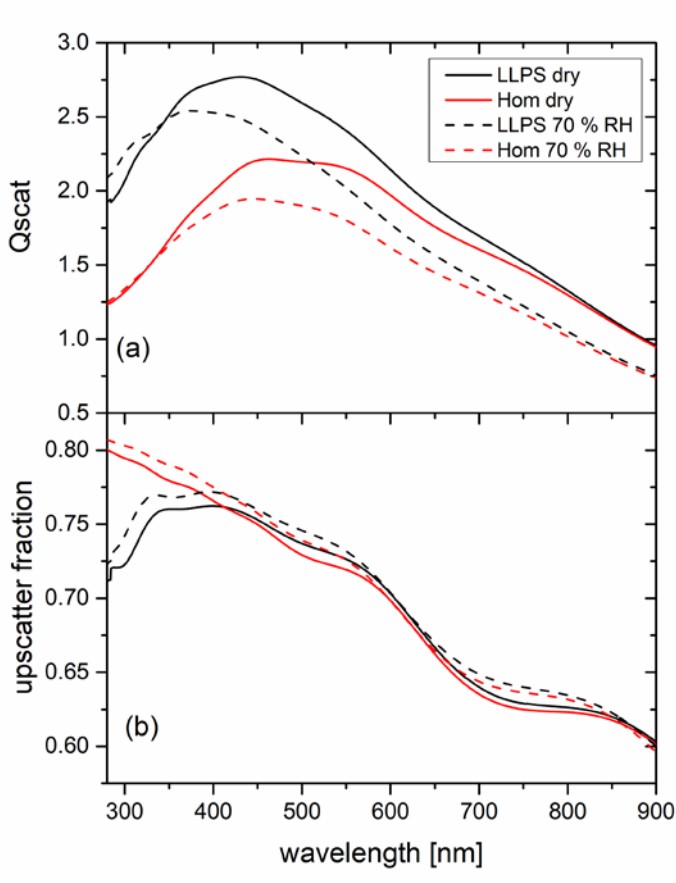

10    **Figure 8 Shown are calculations for the limiting low albedo case. OIR 1:4, diameter 200 nm, $k = 0.168$ at 355 nm. (a): Scattering efficiency for the homogeneous morphology (red) and LLPS morphology (black) under dry and wet conditions (solid and dashed line, respectively) for particles of identical diameter (200 nm) and AAE = 2. (b): Up-scatter fraction for the homogeneous particle (red) and LLPS particle (black) under dry and wet conditions (solid and dashed line, respectively).**

Panel (a) in Fig. 8 shows the scattering efficiency for both, dry conditions and at a relative humidity of 70 %. As discussed above, the LLPS morphology yields larger scattering efficiencies especially at shorter wavelengths at which the differences in refractive indices are more significant. The up-scatter fraction shown in panel (b) for LLPS morphology is about 10 % smaller than for the homogeneous morphology at near UV-wavelength ($\lambda$ = 290 nm) but they merge for the wavelengths above 400 nm.

For calculating the net ratio in radiative forcing of phase-separated particles relative to homogeneously mixed ones, we utilize Eq. (3). Here, the product of up-scatter fraction and scattering efficiency integrated over the short wave solar spectrum for both, LLPS morphology and homogeneous morphology, yields the net ratio that quantifies the effect of morphology on direct radiative forcing. For the solar spectrum we used the spectral irradiance according to ASTM G173-03 (ASTM, 2012) and integrated Eq. (3) from 290 nm to 900 nm, see Appendix C.

The ratio is shown as a function of particle radius under dry and wet (70 % RH) conditions in Figs. 9(a) and 9(b), respectively.

These calculations were done as in the example of Fig. 7 but for different scenarios with OIR = 1:4, 1:1, 4:1, $k$ = 0, 0.1, and 0.168.

Figure 9 shows the results for the case where AAE is equal to 2. This corresponds to highly absorbing BrC and will give the largest radiative forcing impact possible by mixed BrC particles. Figure 10 depicts the result for a less strongly absorbing BrC in the visible range of the solar spectrum, where AAE is chosen to be equal 6.

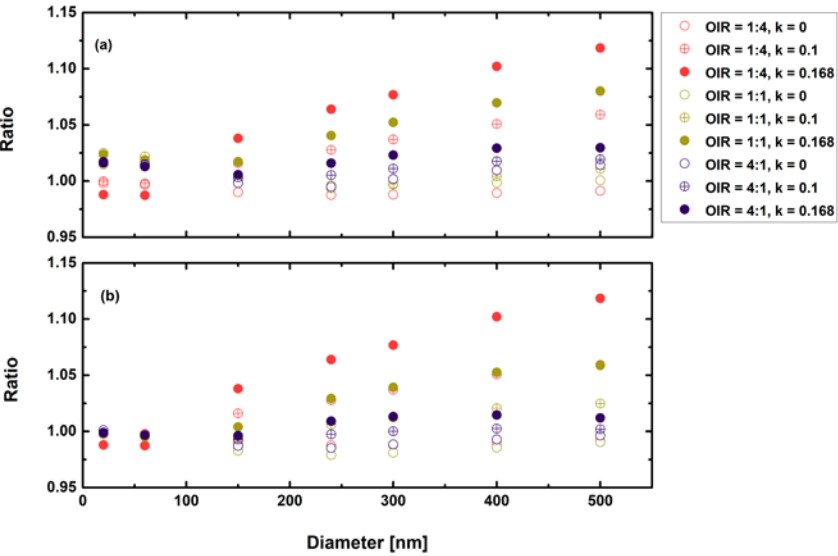

**Figure 9: Ratio of radiative forcing of LLPS to homogenous case under 70 % RH (a) and dry condition (0 % RH). (b) both for AAE = 2 and albedo $a$ = 0.**

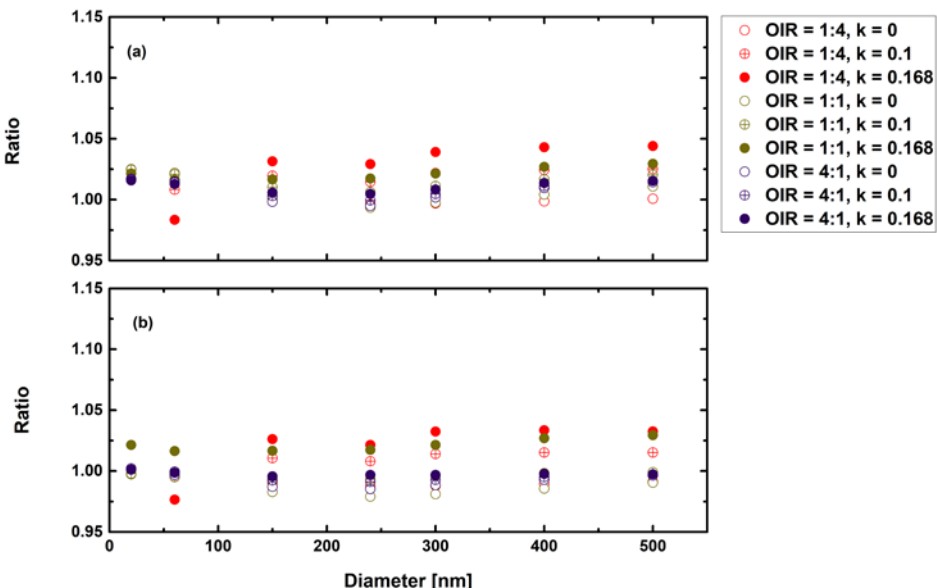

**Figure 10: Ratio of radiative forcing of LLPS to homogenous case under 70 % RH (a) and dry condition (b) for AAE = 6 and albedo *a* = 0.**

First, we conclude from these calculations that the effect of morphology for purely scattering aerosol is negligible, smaller than 2 % for all sizes and organic to inorganic ratios. Second, there is not much difference between dry and moderately humid conditions (remember that at high RH (beyond SRH) we expect the particle to be homogeneously mixed). Third, as expected from the results discussed in the previous section, the greatest effect is calculated when the organic fraction is the lowest (OIR = 1:4), *k* has the largest value (0.168) and the size is on the upper size range of the accumulation mode. However, even here the increase is only about 12 %. For an AAE more likely to occur in aged aerosol, i.e. AAE = 6, this increase reduces to 4 %. Based on the results shown in Figs 9-10, the impact for cases where AAE is lower than 6 is negligible. Since even an AAE of 6 is considered to be characteristic of a strongly absorbing brown carbon, our overall conclusion is that liquid-liquid phase separation has no significant effect on direct short-wave aerosol forcing for low albedos.

Second, we may discuss in a similar manner the high albedo limit, i.e. *a* = 1. Fig. 11 and 12 show the corresponding results.

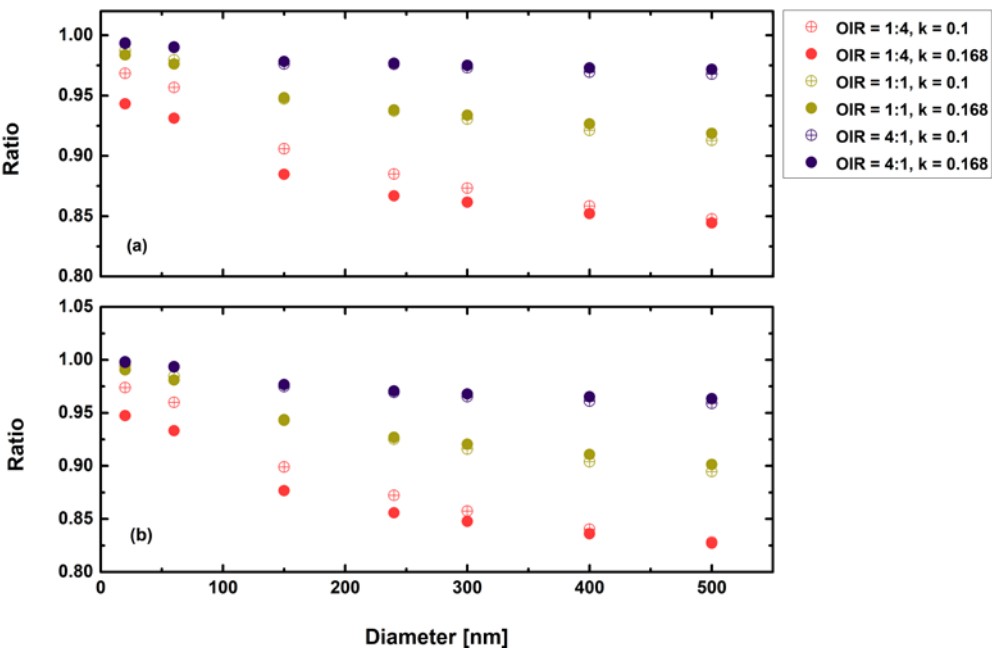

**Figure 11: Ratio of radiative forcing of LLPS to homogenous case under 70 % RH (a) and dry condition (0 % RH). (b) both for AAE = 2 and albedo *a* = 1.**

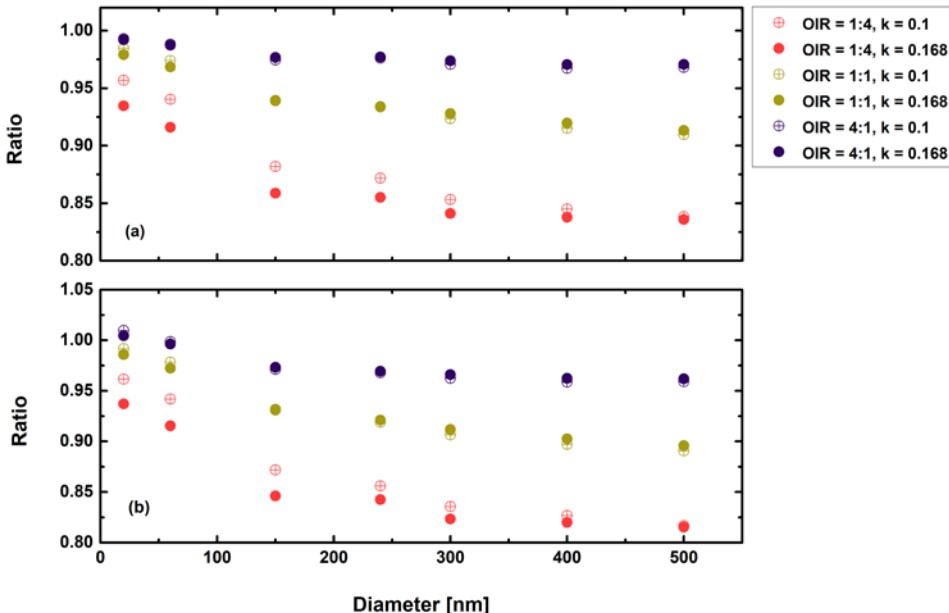

**Figure 12: Ratio of radiative forcing of LLPS to homogenous case under 70 % RH (a) and dry condition (b) for AAE = 6 and albedo *a* = 1.**

5      Again, there are only small differences when comparing the humid and dry cases as well as between the AAE = 6 and AAE = 2 cases. However, the LLPS morphology shows a smaller forcing compared to the homogeneous morphology because $Q_{abs}$ is the decisive parameter for a highly refractive surface (compare Eq. (3) and Fig. 6). Overall, the maximum reduction is 20% for the largest sizes considered here and the OIR equal 1:4 as expected from the discussion above.

10     Up to here, we did only compare ratios for the different morphologies. For a surface albedo close to zero, radiative forcing will be negative for a thin aerosol layer, whereas the forcing will turn positive for a highly reflecting surface for an absorbing aerosol. For intermediate albedos, the denominator of Eq. (3) (the forcing for the homogeneous morphology) will approach zero for a particular size and albedo combination, meaning that the effect of scattering and absorption at this surface albedo cancel out yielding a zero forcing. However, since the corresponding particles with LLPS morphology have a small but finite

15     forcing it results in a very large ratio of the short wave radiative forcing for LLPS to homogenous morphology. This is illustrated in Fig. 13.

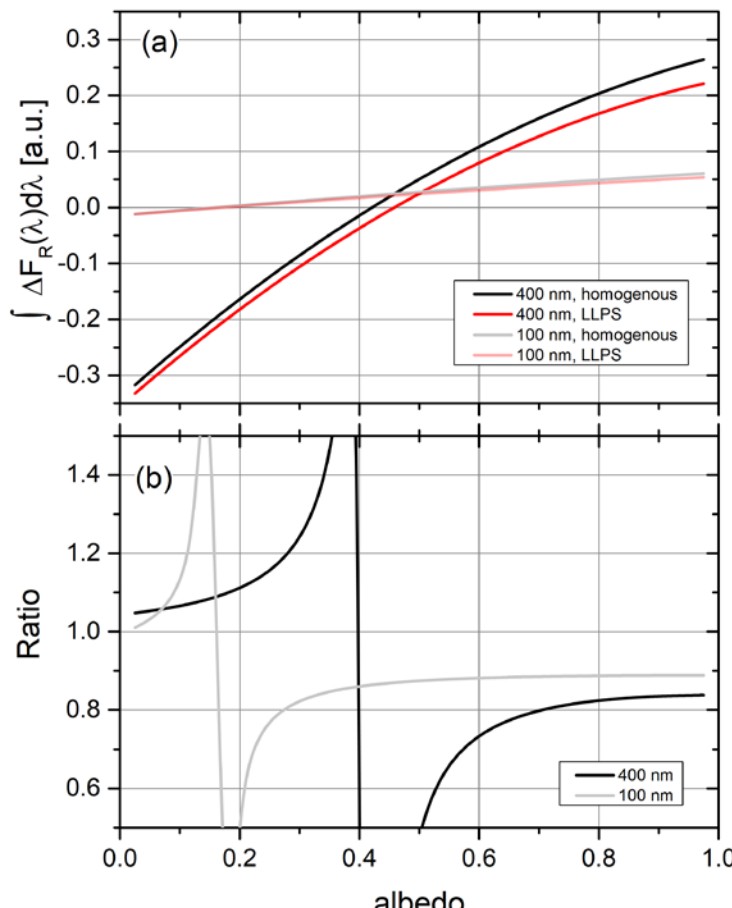

**Figure 13 (a): Direct radiative forcing integrated over the visible solar spectrum for particles with OIR 1:4, AAE = 6, and k = 0.168 at 355 nm. Results for two diameters are shown. (b) Ratio of forcing for LLPS morphology to homogeneous morphology, see**

5    **Eq. (3), for the data of (a).**

Panel (a) shows clearly, that the albedo for which the direct radiative forcing vanishes, depend on the size of the particle, shifting to larger albedos with increasing particle size. This leads to poles in the ratio of forcing for the two morphologies as seen in panel (b) of Fig. 12. However, for a more realistic atmospheric situation where the thin aerosol layer will contain

10   particles with sizes and refractive indices distributed over a significant range these poles will level out. Hence, we expect a

smooth transition for the ratio of radiative with a larger negative forcing for LLPS morphology at low albedos to a smaller positive forcing at high albedos for LLPS morphology compared to homogeneous morphology.

**4 Conclusions**

Using both eccentric and concentric core-shell model calculations for scattering and absorption efficiencies of single aerosol particles with an inorganic non-absorbing core and an absorbing organic shell at different volume ratios, sizes and absorptivity revealed that a concentric core-shell model is a good approximation for calculating these efficiencies. Applied to liquid-liquid phase separation for atmospheric relevant OIR and sizes typical of the accumulation mode we showed that the largest impact resulted from the case where organic fraction had the lowest contribution in the mixed particle (OIR = 1:4) and formed a very thin shell around the inorganic core and was highly absorbing ($k = 0.168$). Once integrated over the solar spectrum, taking into account the typical spectral dependence of BrC, the effect of morphology on radiative forcing substantially decreased to about a few percent. Overall, we conclude that the effect of liquid-liquid phase separation on short wave radiative forcing is rather small and the correct value of AAE is the greatest source of uncertainty when estimating for the impact.

# Appendices

## Appendix A. Calculating refractive indices for homogenous and phase-separated particles

Tables A1 and A2 show the values for the refractive indices, $n + i\,k$, at selected imaginary refractive indices ($k = 0.168$ in Table A1 and $k = 0.01$ in Table A2) set for the homogenous particle. The corresponding size of the core in the LLPS morphology as well as the refractive indices were calculated using simple volume mixing. These values are used as inputs for the calculations shown in Fig. 6.

**Table A1: Calculated values for relative size of the core to shell for phase-separated particles and refractive indices ($k = 0.168$) for homogenous and LLPS case at different size and OIRs using volume mixing**.

| Particle Size (nm) | OIR | Morphology | $r_{shell}$ (nm) | $r_{core}$ (nm) | $n_{core}$ | $n_{shell}$ | $k_{core}$ | $k_{shell}$ |
|---|---|---|---|---|---|---|---|---|
| 40 | 1:1 | homogeneous | - | | 1.5 | | 0.168 | |
| | | LLPS | 20 | 15.87 | 1.429 | 1.571 | 0 | 0.336 |
| 40 | 1:4 | homogeneous | - | | 1.457 | | 0.168 | |
| | | LLPS | 20 | 18.57 | 1.429 | 1.571 | 0 | 0.84 |
| 40 | 4:1 | homogeneous | - | | 1.543 | | 0.168 | |
| | | LLPS | 20 | 11.69 | 1.429 | 1.571 | 0 | 0.21 |
| 200 | 1:1 | homogeneous | - | - | 1.5 | | 0.168 | |
| | | LLPS | 100 | 79.37 | 1.429 | 1.571 | 0 | 0.336 |
| 200 | 1:4 | homogeneous | - | - | 1.457 | | 0.168 | |
| | | LLPS | 100 | 92.8 | 1.429 | 1.571 | 0 | 0.84 |
| 200 | 4:1 | homogeneous | - | - | 1.543 | | 0.168 | |
| | | LLPS | 100 | 58.9 | 1.429 | 1.571 | 0 | 0.21 |
| 500 | 1:1 | homogeneous | - | - | 1.5 | | 0.168 | |
| | | LLPS | 250 | 198.4 | 1.429 | 1.571 | 0 | 0.336 |
| 500 | 1:4 | homogeneous | - | - | 1.457 | | 0.168 | |
| | | LLPS | 250 | 232 | 1.429 | 1.571 | 0 | 0.84 |
| 500 | 4:1 | homogeneous | - | - | 1.543 | | 0.168 | |
| | | LLPS | 250 | 146.2 | 1.429 | 1.571 | 0 | 0.21 |

**Table A2: Calculated values for relative size of the core to shell for phase-separated particles and refractive indices ($k = 0.01$) for homogenous and LLPS case at different size and OIRs using volume mixing.**

| Particle Size (nm) | OIR | Morphology | $r_{shell}$ (nm) | $r_{core}$ (nm) | $n_{core}$ | $n_{shell}$ | $k_{core}$ (nm) | $k_{shell}$ (nm) |
|---|---|---|---|---|---|---|---|---|
| 40 | 1:1 | homogeneous | - | - | 1.5 | | 0.01 | |
| | | LLPS | 20 | 15.87 | 1.429 | 1.571 | 0 | 0.02 |
| 40 | 1:4 | homogeneous | - | - | 1.457 | | 0.01 | |
| | | LLPS | 20 | 18.57 | 1.429 | 1.571 | 0 | 0.05 |
| 40 | 4:1 | homogeneous | - | - | 1.543 | | 0.01 | |
| | | LLPS | 20 | 11.69 | 1.429 | 1.571 | 0 | 0.0125 |
| 200 | 1:1 | homogeneous | - | - | 1.5 | | 0.01 | |
| | | LLPS | 100 | 79.37 | 1.429 | 1.571 | 0 | 0.02 |
| 150 | 1:4 | homogeneous | - | - | 1.457 | | 0.01 | |
| | | LLPS | 75 | 69.6 | 1.429 | 1.571 | 0 | 0.05 |
| 200 | 4:1 | homogeneous | - | - | 1.543 | | 0.01 | |
| | | LLPS | 100 | 58.9 | 1.429 | 1.571 | 0 | 0.0125 |
| 500 | 1:1 | homogeneous | - | - | 1.5 | | 0.01 | |
| | | LLPS | 250 | 198.4 | 1.429 | 1.571 | 0 | 0.02 |
| 500 | 1:4 | homogeneous | - | - | 1.457 | | 0.01 | |
| | | LLPS | 250 | 232 | 1.429 | 1.571 | 0 | 0.05 |
| 500 | 4:1 | homogeneous | - | - | 1.543 | | 0.01 | |
| | | LLPS | 250 | 146.2 | 1.429 | 1.571 | 0 | 0.0125 |

**Appendix B. Estimating the refractive index for the calculations of section 3.**

The real part of the refractive index for a liquid solution may be estimated in terms of the refractivity of the solution based on the Lorentz-Lorenz relation (Born & Wolf, 1959). The refractivity, to a good approximation, is a linear superposition of the molar refractivities of the solution's components. While refractive index data as well as density data are available for aqueous ammonium sulfate (AS) solutions (Tang & Munkelwitz, 1994), we choose the refractive index and density parameterizations of Lienhard et al. (2015) to be representative for the secondary organic matter (SOM) in our model calculations. As the molar refractivities depend strongly on wavelength, we parameterize the SOM molar refractivity wavelength dependence based on the parameterization given in Liu et al. (2013) and the ones for aqueous ammonium sulfate on the parameterization by Semmler et al., (2018). Finally, we use ideal mixing of the two binary systems to calculate the refractive index of the ternary system. The resulting refractive indices for the ternary system with different OIR under dry conditions are shown in Fig. B1.

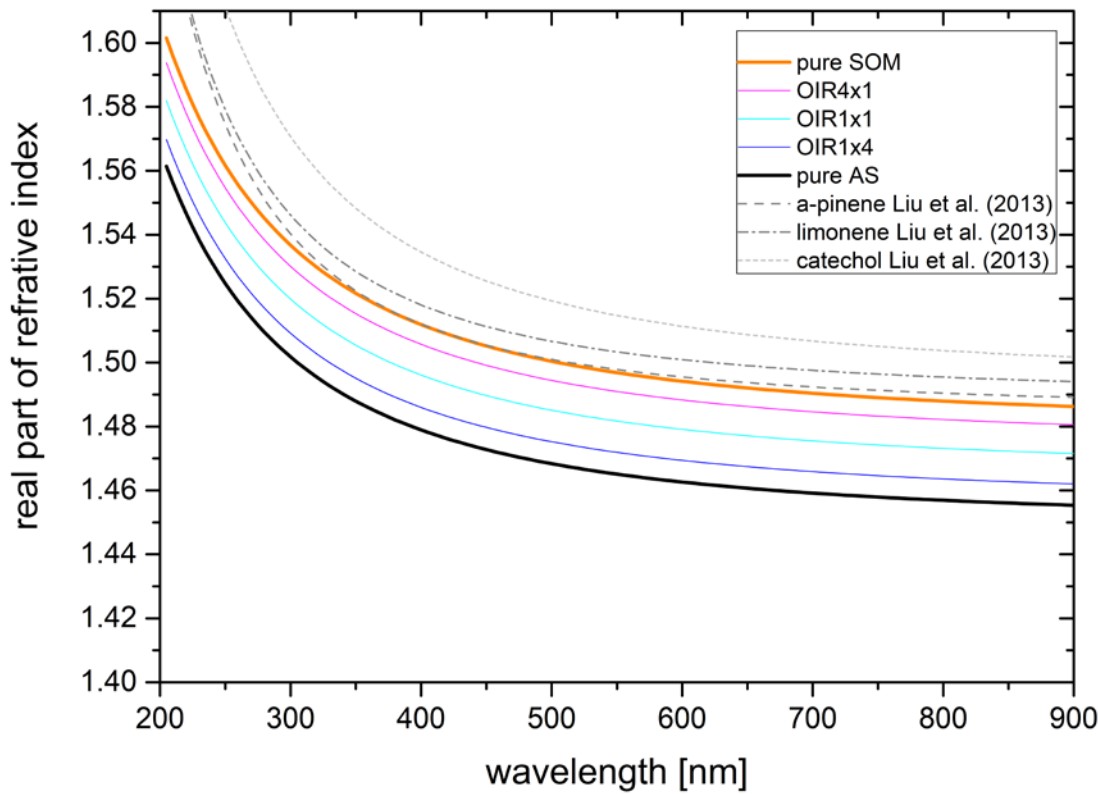

**Figure B1: Real part of refractive index, *n*, for aqueous mixtures of ammonium sulfate (AS) and secondary organic matter (SOM) with varying OIR extrapolated to dry condition (lines in various colors). For comparison, the parametrizations of Liu et al. (2013)**

5    **for SOM obtained by ozonolysis of α-pinene, limonene and catechol are given (gray lines).**

To calculate the refractive indices at 70 % relative humidity we use the water activity of the binary aqueous solutions for AS (Tang & Munkelwitz, 1994) and SOM (Lienhard et al. 2015), and the Zdanovskii-Stokes-Robinson (ZSR) relation to

10    calculate the water content of the AS-SOM mixture. This yields the refractive indices shown in Fig. B2.

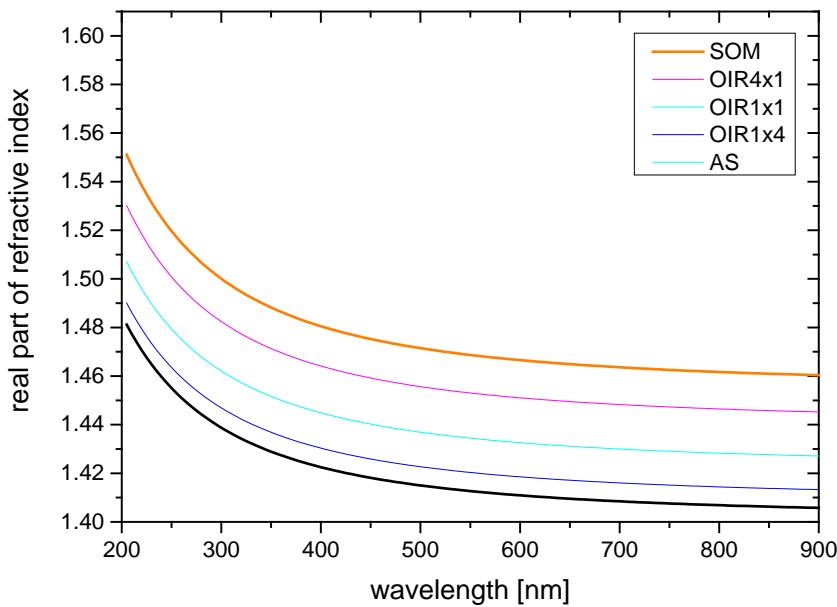

**Figure B2: Real part of refractive index for mixtures of AS and SOM with varying OIR at 70 % RH.**

Under humid conditions the real part of the refractive index decrease and, since AS takes up more water at 70 % compared to SOM, the difference in refractive index between AS rich mixtures to SOM rich mixtures increases when comparing humid to dry conditions.

The wavelength dependence of the imaginary part of the refractive index is taken into account by assuming the simple power law dependence of Eq. (4). In Fig. B3 we show the imaginary part of the refractive index as a function of wavelength for two Ångström exponents, with the k = 0.168 at $\lambda$ = 355 nm. For comparison, we plot the parameterizations used by Wang et al. (2014) and the data collected in this reference as well.

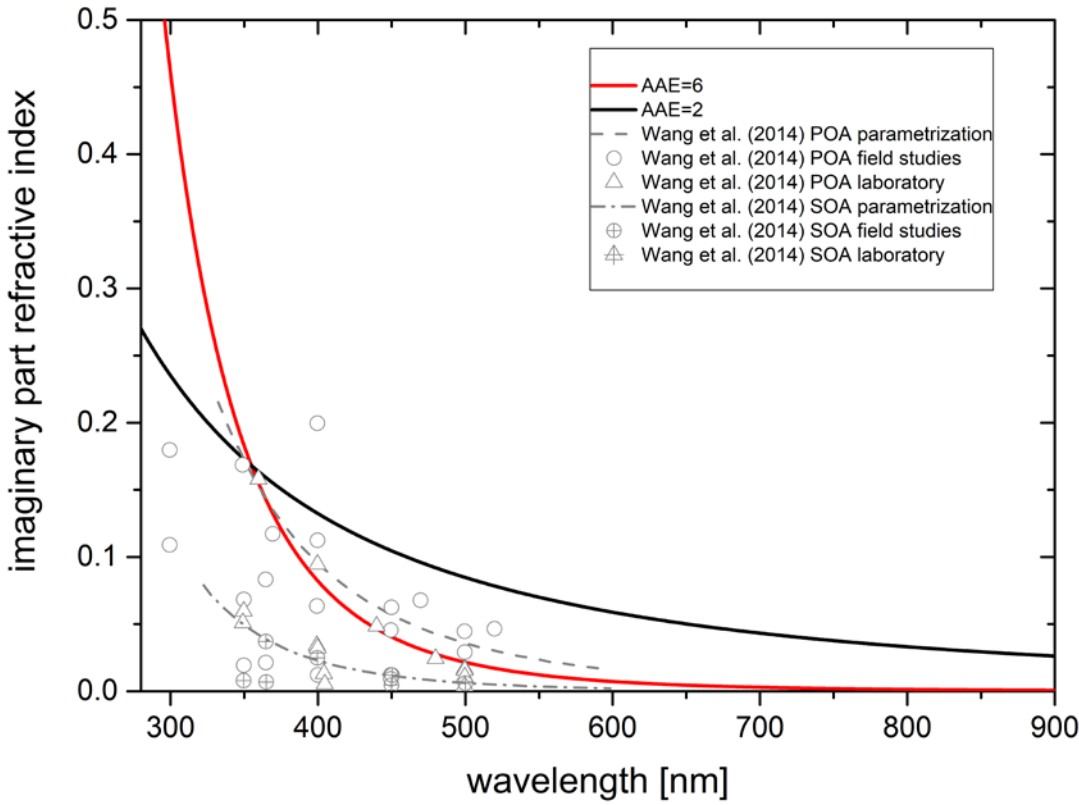

**Figure B3: Wavelength dependence of the imaginary part of the refractive index for AAE equal to 2 and 6 (solid black and red lines, respectively. $k = 0.168$ for $\lambda = 355$ nm. For comparison the parametrizations of Wang et al. (2014) for brown primary organic aerosol (POA, dashed gray line) and brown secondary organic aerosol (SOA, dashed-dotted gray line) are plotted as well as the data from laboratory and field studies collected by Wang et al. (2014).**

Clearly, the AAE=2 case poses an upper limit of absorptivity, whereas the AAE=6 case is in-between of the parametrization for brown carbon primary organic aerosol and brown carbon secondary aerosol estimates of Wang et al. (2015).

### Appendix C. Spectral Irradiance

For calculating the shortwave radiative forcing ratio defined in Eq. (3), spectral irradiance is needed as an input for performing the integration. Since we are interested in estimating the relevance of LLPS for radiative forcing and calculating only a ratio, the particular choice of irradiance data is not very important. We use the ASTM G173-03 (ASTM, 2012) as spectral irradiance, which is plotted in Fig. C1.

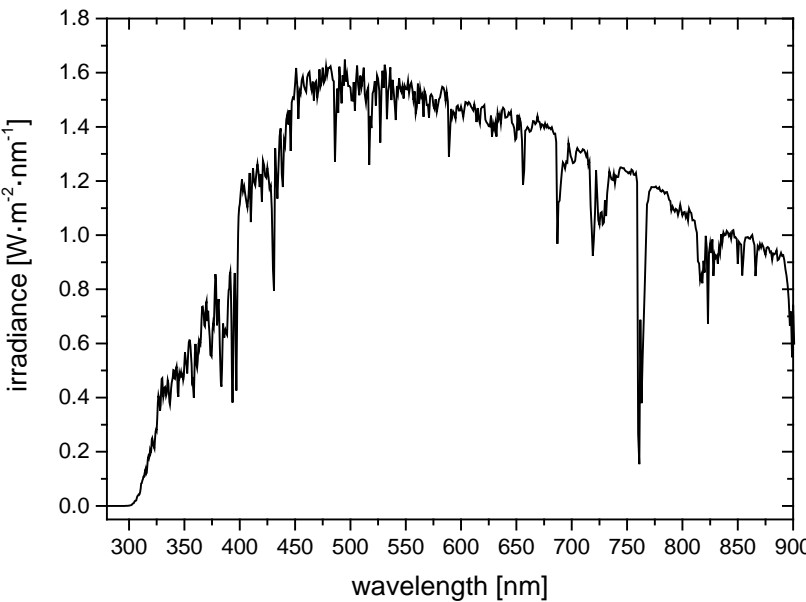

**Figure C1: Direct spectral irradiance (ASTM G173-03).**

The irradiance is for a solar zenith angle of 41.81°, the atmospheric conditions are those of the standard US atmosphere with

5   an ozone column of 340 DU and total column water vapor equivalent of 1.42 cm.

**Acknowledgements**

We would like to thank Dan Mackowski for making his MSTM code publicly available and an anonymous reviewer for insightful comments and suggestions. This project was funded by the Swiss National Science Foundation (SNF, Grant No.

10   200020 146760/1).

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
