# Peer review of "Short wave radiative impact of liquid-liquid phase separation in "Brown Carbon" aerosols"

_Atmospheric Chemistry and Physics, 2018_

## Referee Comment (RC1) · Anonymous Referee #1 · 25 Apr 2018

Review of "Short wave radiative impact of liquid-liquid phase separation . . ." by Fard et al.

This manuscript investigates the impact of liquid-liquid phase separation in particles containing organic and inorganic material on scattering efficiencies, absorption efficiencies, and radiative forcing. Although liquid-liquid phase separation has been studied extensively, the impact of this phase transition on radiative forcing of particles containing molecular absorbers has not been studied until now. As a result, this manuscript is important and timely. I highly recommend this manuscript for publication after the authors have had a chance to address the following comments.

1) Page 4. Check spelling of "Zhnag".

[Figure]

2) Page 4, line 10. The authors indicate that the k-values correspond to SOA from different locations. Does this include brown carbon from biomass burning and smoldering combustion? Do the k-values used in the simulations cover the full range of k-values observed in the atmosphere? Although not absolutely necessary, it would be very helpful if the authors discussed k-values and AAE–values corresponding to different types of brown carbon found in the atmosphere. For example, what are typical values for biomass burning, smoldering combustion, SOA generated in environmental chambers, and organic material collected in the atmosphere? A small table would be very helpful. This would make it easier for a non-expert to put the results into context.

3) Figure 3. What is plotted on the x-axis (include units)?

4) Page 10, line 15, delete "with".

5) Equation 4. On the denominator should "betaLLPS" be replaced with "betaHomo"?

6) AAE values ranging from 2 to 6 where used. References for these values should be included. Sorry if I missed the references.

7) Figure B1. Shown is the refractive indices for pure SOM from Lienhard. What type of SOM (e.g. pinene or toluene SOM) was used to determine these refractive indices? Also how do these refractive indices compare with what is observed in atmospheric particles? Will the authors reach different conclusions if a different type of SOM is used?

---

## Referee Comment (RC2) · Anonymous Referee #2 · 25 Apr 2018

This is a very interesting and useful manuscript that should be suitable for publication in ACP after the flawed discussion of direct radiative forcing in section 3 (Atmospheric Implications) has been redone correctly. Detailed comments are as follows:

P1L29-31: Add the reference of Bond et al. (2013) and change the sentence also referring to the surface albedo (e.g., Chylek and Wong, 1995) and changing the unclear wording "may also contribute to warming". Does this mean that in addition to cooling they also cause warming?

P3L9: "Brown Carbon is referring to the light-absorbing fraction of the organic carbon that has a wavelength dependent absorptivity." This is a very poor definition of BrC because the key definition is that the imaginary part of the refractive index (not the absorptivity) is wavelength dependent and increases toward shorter wavelengths (e.g.,

[Figure]

Moosmuller et al., 2011).

P3L13: "We use simple volume mixing...". This needs some explanation of effective medium theories, why the volume mixing rule was chosen, and what its accuracy is. A good starting point would be Chylek et al. (1988).

P7 Fig. 2: (1) Explain exactly what is meant here with random location and how it is realized computationally; (2) Give the complex refractive index both for the particle and the inclusion here and elsewhere; (3) "100 nm particle": Does "100 nm" refer to particle radius, diameter, circumference or something else; please state explicitly here and elsewhere!

P12L22 – P17L12: "3 Atmospheric Implications". This section is flawed and in need of major revision! The reference Charlson et al. (1991) discusses only radiative forcing by non-absorbing (i.e., sulfate) aerosols; the reference Nemesure and Schwartz (1998) is in the "grey" literature and should be replace with a peer-reviewed reference such as Chylek and Wong (1995). In addition, the authors pick the wrong equation from Nemesure and Schwartz (1998) that doesn't account for the albedo of the underlying surface. In reality, the radiative forcing in the optically thin aerosol layer case depends on one extensive aerosol parameter (AOD), two intensive aerosol parameters (SSA and upscatter fraction), and the albedo of the underlying surface or scene. The equation for this can be found in Nemesure and Schwartz (1998) p. 532, left column just above the right column header "Results" or in the peer reviewed literature (Chylek and Wong; 1995; eq. 8), with further discussion of validity and assumptions to be found in Hassan et al. (2015), Moosmuller and Ogren (2017), and Moosmuller and Sorensen (2018). Of specific interest would be to plot the ratio (LLPS/homogeneous) of the dominating intensive aerosol forcing parameter SSA as function of particle diameter such as done in Fig. 6 for Qscat and Qabs.

References

Bond, T. C., S. J. Doherty, D. W. Fahey, P. M. Forster, T. Berntsen, B. J. DeAngelo, M. G.

[Figure]

Flanner, S. Ghan, B. Kärcher, D. Koch, S. Kinne, Y. Kondo, P. K. Quinn, M. C. Sarofim, M. G. Schultz, M. Schulz, C. Venkataraman, H. Zhang, S. Zhang, N. Bellouin, S. K. Guttikunda, P. K. Hopke, M. Z. Jacobson, J. W. Kaiser, Z. Klimont, U. Lohmann, J. P. Schwarz, D. Shindell, T. Storelvmo, S. G. Warren, and C. S. Zender (2013). Bounding the Role of Black Carbon in the Climate System: A Scientific Assessment. J. Geophys. Res., 118, 5380-5552.

Chylek, P., V. Srivastava, R. G. Pinnick, and R. T. Wang (1988). Scattering of Electromagnetic Waves by Composite Spherical Particles: Experiment and Effective Medium Approximations. Appl. Opt., 27, 2396-2404.

Chylek, P., and J. Wong (1995). Effect of Absorbing Aerosol on Global Radiation Budget. Geophys. Res. Lett., 22, 929-931.

Hassan, T., H. Moosmuller, and C. E. Chung (2015). Coefficients of an Analytical Aerosol Forcing Equation Determined with a Monte-Carlo Radiation Model. J. Quant. Spectrosc. Radiat. Transfer, 164, 129-136.

Moosmuller, H., R. K. Chakrabarty, K. M. Ehlers, and W. P. Arnott (2011). Absorption Ångström Coefficient, Brown Carbon, and Aerosols: Basic Concepts, Bulk Matter, and Spherical Particles. Atmos. Chem. Phys., 11, 1217-1225.

Moosmuller, H., and J. A. Ogren (2017). Parameterization of the Aerosol Upscatter Fraction as Function of the Backscatter Fraction and Their Relationships to the Asymmetry Parameter for Radiative Transfer Calculations. Atmosphere, 8, doi:10.3390/atmos8080133.

Moosmuller, H., and C. M. Sorensen (2018). Small and Large Particle Limits of Single Scattering Albedo for Homogeneous, Spherical Particles. J. Quant. Spectrosc. Radiat. Transfer, 204, 250-255.

---

## Author Comment (AC1) · 8 Aug 2018

We thank reviewer 1 for her/his assessment of our paper and his/her comments and suggestions, which we will reply to point by point below.

**1) Page 4.** *Check spelling of "Zhnag".*

Typo.

**Changes to the manuscript:** Typo will be corrected. In addition, this reference was missing in the list of references and will be added.

**2) Page 4, line 10.** *The authors indicate that the k-values correspond to SOA from different locations. Does this include brown carbon from biomass burning and smoldering combustion? Do the k-values used in the simulations cover the full range of k-values observed in the atmosphere? Although not absolutely necessary, it would be very helpful if the authors discussed k-values and AAE-values corresponding to different types of brown carbon found in the atmosphere. For example, what are typical values for biomass burning, smoldering combustion, SOA generated in environmental chambers, and organic material collected in the atmosphere? A small table would be very helpful. This would make it easier for a non-expert to put the results into context.*

We thank the reviewer for the suggestion. Although additional data became available, we believe the ranges given in Table 3 of Moise at al. (2015) are still representative for the k-values of the different sources. Rather than providing any mean values, the ranges reported at about 355 nm taken from this table are:

Laboratory reacted organic compounds
(biogenic SOA) 9E-4 to 3.7E-3
(anthropogenic SOA) 4.7E-2
(HULIS proxies) 4.6E-2 to 9.8E-2
(ammonia mediated aging of SOA) 7E-3 to 3.1E-2

Ambient aerosol
(pollution Hulis) 9.8E-2
(smoke HULIS) 1.16E-1
(rural HULIS) 2.3E-2
(biomass burning HULIS) 7E-3

Taking this compilation, it is evident, that our k-values cover the full range of the atmospherically relevant values, with k=0.168 @355 nm being one of the largest k-values observed. You may also look at Fig. 1 of Wang et al. (2014), whose data we added to the revised Fig. B3, see below.

**Changes to the manuscript:** We will add the Moise at al. (2015)

and Wang et al. (2014) references and change the sentence
starting at line 10 to:
"To account for the absorptivity of BrC, we take the imaginary
parts of the refractive index (k) for BrC spanning a wide
range from non-absorbing organic material (k = 0) to highly
absorbing organic matter (k = 0.168 at 355 nm). This range is
based on various studies (Kirchstetter et al., 2004; Chen and
Bond, 2010; Feng et al., 2013, Wang et al., 2014, Moise et
al., 2015) that measured or collected data of k for different
absorbing aerosol at different locations."

**3) Figure 3.** *What is plotted on the x-axis (include units)?*

We thank the reviewer for pointing out the missing
description. Plotted is scattering efficiency versus the
position of the center of the core relative to the center of
the particle. Units of the original figure were μm.

**Changes to the manuscript:** We will change the axis title of
Fig. 3 and its caption as well. New caption:
"Figure 3. The change in Qscat with the relative position of
the core (r(core) = 92.8 nm) to the particle center
perpendicular to the direction of light, x- and y-axis (panel
a) and along the direction of light , z-axis (panel b) for
particle with OIR = 1:4, k = 0.168, r(particle) = 100 nm over
10000 realizations."

**4) Page 10, line 15,** *delete "with".*

**Changes to the manuscript:** will delete.

**5) Equation 4.** *On the denominator, should "betaLLPS" be
replaced with "betaHomo"?*

Yes, the reviewer is correct. However, we will change the
whole section following the advice of reviewer 2, see the
answers to the comments of reviewer 2.

**Changes to the manuscript:** revised version of the atmospheric
implication section

**6)** *AAE values ranging from 2 to 6 where used. References for
these values should be included. Sorry if I missed the
references.*

We take the advice of the reviewer and will put more detailed
information into Fig. B3 from Wang et al. (2014).

**Changes to the manuscript:** We will revise Fig. B3 by adding the parametrizations of Wang et al. (2014) for comparison as well as the data collected in this reference. We will add a sentence to the text of Fig. B3: "Clearly, the AAE=2 case poses an upper limit of absorptivity, whereas the AAE=6 case is in-between of the parametrization for brown carbon primary organic aerosol and brown carbon secondary aerosol estimates of Wang et al. (2015)."

**7) Figure B1**. *Shown is the refractive indices for pure SOM from Lienhard. What type of SOM (e.g. pinene or toluene SOM) was used to determine these refractive indices? Also how do these refractive indices compare with what is observed in atmospheric particles? Will the authors reach different conclusions if a different type of SOM is used?*

The SOM of Lienhard et al. (2015) was generated in a PAM chamber by OH oxidation of α-pinene. The real part of refractive index is almost identical to the one determined by Liu et al. (2013) for particles generated by ozonolysis of alpha-pinene. Liu et al. (2013) measure the one for limonene and catechol as well, with catechol having a larger index compared to our SOM (catechol @ 550 nm: 1.5147, our SOM: 1.4968).
However, for the simulations shown in Fig. 6 we used an even higher real part of the refractive index than that of catechol and do not see any significant differences compared to the simulations shown in Figs 8 and 9. Therefore, we conclude that the exact value for the real part of the refractive index will not lead to different conclusions.

**Changes to the manuscript:**

We will add the parametrizations of Liu et al. (2013) to Fig. B1 to allow a comparison.

**Revised figures:**

[Figure]

**Figure B1: Real part of refractive index, *n*, for aqueous mixtures of ammonium sulfate (AS) and secondary organic matter (SOM) with varying OIR extrapolated to dry condition (lines in various colors). For comparison, the parametrizations of Liu et al. (2013) for SOM obtained by ozonolysis of α-pinene, limonene and catechol are given (gray lines).**

[Figure]

**Figure B3: Wavelength dependence of the imaginary part of the refractive index for AAE equal to 2 and 6 (solid black**

and red lines, respectively. $k = 0.168$ for $\lambda = 355$ nm. For comparison the parametrizations of Wang et al. (2014) for brown primary organic aerosol (POA, dashed gray line) and brown secondary organic aerosol (SOA, dashed-dotted gray line) are plotted as well as the data from laboratory and field studies collected by Wang et al. (2014).

---

## Author Comment (AC2) · 8 Aug 2018

We thank reviewer 2 for her/his comments and suggestions, in particular, we are grateful for her/him pointing out our neglect of surface albedo in the atmospheric implication section. We will reply to point by point below.

**P1L29-31:** Add the reference of Bond et al. (2013) and change the sentence also referring to the surface albedo (e.g., Chylek and Wong, 1995) and changing the unclear wording "may also contribute to warming". Does this mean that in addition to cooling they also cause warming?

We agree with the reviewer that surface albedo is very important for evaluating whether an aerosol is heating of cooling. We will add this to the sentence.
However, while purely scattering aerosol particles will cool the surface, strong absorbing aerosol can heat the planet. The sentence reads in the manuscript: "Depending on their optical properties, aerosols contribute mostly to the cooling of our planet (IPCC, 2013) but when they are highly absorptive (e.g., soot) may also contribute to warming (e.g. Ramanathan et al., 2001)." That was not meant "in addition", but "instead". We will rephrase the sentence.

**Changes to the manuscript:** We will rephrase the sentence to: "Depending on their optical properties, size and albedo of the surface, aerosols mostly cool our planet (IPCC, 2013). However, those which are highly absorptive (e.g., soot particles) can lead to heating (e.g. Ramanathan et al., 2001, Bond et al., 2013). We will add the reference Bond et al. (2013).

**P3L9:** "Brown Carbon is referring to the light-absorbing fraction of the organic carbon that has a wavelength dependent absorptivity." This is a very poor definition of BrC because the key definition is that the imaginary part of the refractive index (not the absorptivity) is wavelength dependent and increases toward shorter wavelengths (e.g., Moosmuller et al., 2011).

We will take the advice of the reviewer and will change the sentence. However, while we agree that the sentence is a bit misleading with brown carbon being no single compound, it is clear that if nevertheless approximating it as such, its molar absorptivity would be indeed wavelength dependent, as the absorption coefficient is directly proportional to the imaginary part of the refractive index.

**Changes to the manuscript:** Sentence will be revised to: "Brown Carbon is referring to the light-absorbing fraction of the organic carbon that has a wavelength dependent imaginary part

of the refractive index, which increases towards shorter wavelengths"
**P4L13:** "We use simple volume mixing. . .". This needs some explanation of effective medium theories, why the volume mixing rule was chosen, and what its accuracy is. A good starting point would be Chylek et al. (1988).

In this section of the manuscript, we just want to show that the concentric core-shell model is a good approximation for calculating the mean value for a distribution of particles with randomly located eccentric cores. While the magnitude of the calculated efficiencies depend strongly on the real part of the refractive index, the core shell model is always a good for various assumed refractive indices. Hence we used simple volume mixing here (in contrast to what we do in appendix B, for estimating the atmospheric implications), just for illustration.

**Changes to the manuscript:** We will add a sentence stating this after line 15: (Note, we use the volume mixing approximation just to illustrate the effect of morphology in this section, for this purpose any effective medium approximation could be used.)

**P7 Fig. 2:** (1) Explain exactly what is meant here with random location and how it is realized computationally; (2) Give the complex refractive index both for the particle and the inclusion here and elsewhere; (3) "100 nm particle": Does "100 nm" refer to particle radius, diameter, circumference or something else; please state explicitly here and elsewhere!

Answer to (1). After LLPS there is only a certain volume accessible for the spherical core if we assume core-shell morphology, i.e. that the core is completely embedded in the spherical particle. We did two types of randomized calculations for the position of the core within the shell: (1) if the core remains always attached to the inside surface of the particle, the radial distance between center of core and center of the particle remains fixed and we used a random number generator to draw random numbers for both, the polar and the azimuthal angle to place the core within the particle in a spherical coordinate system. (The light is always parallel to the z-axis.) (2) If the core is not attached, we also varied the distance between core center and particle center, i.e. the radial coordinate in the spherical coordinate system, by using a random number scaled such that the core access the volume within the particle with equal probability.

Answer to (2). These are given in Table A1. However, we agree that it is helpful to have those in the figure captions.

Answer to (3). We use always diameter, when writing about the size of the particle in the text.

**Changes to the manuscript:** We will add the explanation given in (2) to the text in page 6 line 7.
"We did two types of randomized calculations for the position of the core within the shell. (1) Random position attached to inner surface: the core remains always attached to the inside surface of the particle, hence in a spherical coordinate system the radial distance between center of core and center of the particle remains fixed. We used a random number generator to draw random numbers for both, the polar and the azimuthal angle to place the core within the particle in the spherical coordinate system. The light is always parallel to the z-axis of a corresponding Cartesian coordinate system. (2) Random position within the volume: if the core is not attached, we also varied the distance between core center and particle center, i.e. the radial coordinate in the spherical coordinate system, by using a random number scaled such that the core access the volume within the particle with equal probability."

**P12L22 – P17L12:** "3 Atmospheric Implications". This section is flawed and in need of major revision! The reference Charlson et al. (1991) discusses only radiative forcing by non-absorbing (i.e., sulfate) aerosols; the reference Nemesure and Schwartz (1998) is in the "grey" literature and should be replace with a peer-reviewed reference such as Chylek and Wong (1995). In addition, the authors pick the wrong equation fromNemesure and Schwartz (1998) that doesn't account for the albedo of the underlying surface. In reality, the radiative forcing in the optically thin aerosol layer case depends on one extensive aerosol parameter (AOD), two intensive aerosol parameters (SSA and upscatter fraction), and the albedo of the underlying surface or scene. The equation for this can be found in Nemesure and Schwartz (1998) p. 532, left column just above the right column header "Results" or in the peer reviewed literature (Chylek and Wong; 1995; eq. 8), with further discussion of validity and assumptions to be found in Hassan et al. (2015), Moosmuller and Ogren (2017), and Moosmuller and Sorensen (2018). Of specific interest would be to plot the ratio (LLPS/homogeneous) of the dominating intensive aerosol forcing parameter SSA as function of particle diameter such as done in Fig. 6 for Qscat and Qabs.

Again, we would like to thank the reviewer very much for pointing this out. Our data of Fig. 8 and Fig. 9 are calculations for the albedo being 0, i.e. a completely absorbing surface. We will add corresponding figures for the case of a perfectly reflecting surface as well as a figure showing the effect of surface albedo on the ratio of LLPS to

homogenous forcing for the OIR, size and k with the strongest
overall effect. We will also follow the suggestion to plot the
ratio of SSA for the two morphologies and will use this
additional figure to start the discussion in the atmospheric
implications section.

**Changes to the manuscript:** Since there will be considerable
changes for this section, we do not list all changes here, but
refer to the completely revised atmospheric implications
section. In addition, the last sentence of the abstract will
be modified to reflect these changes to:

[revised manuscript text omitted]

---

## Author Response (AR2)

**Technical corrections:**

**Co-Editor Decision: Publish subject to technical corrections** (30 Aug 2018) by Sergey A. Nizkorodov
Comments to the Author:
Please address the remaining minor points raised by the reviewer (reproduced below).

"The resubmission of this manuscript is much improved and it should be published in ACP after a few
minor technical corrections as follows:
1. P3L9 The authors still argue that "the absorption coefficient is directly proportional to the
imaginary part of the refractive index." While this is true for bulk material and for very small
particles, it is not true in general, see Moosmuller et al., 2011.
2. A couple of the references that I suggested: P31L15: "Chylek" should read "Chýlek", p.33L17:
"Mossmüller" should read "Moosmüller".
References:
Moosmuller, H., R. K. Chakrabarty, K. M. Ehlers, and W. P. Arnott. 2011. "Absorption Ångström
Coefficient, Brown Carbon, and Aerosols: Basic Concepts, Bulk Matter, and Spherical Particles."
Atmospheric Chemistry and Physics 11:1217-1225."

Reply to 1.: This was already changed in the revised version to: "Brown Carbon is referring to the
light-absorbing fraction of the organic carbon that has a wavelength dependent imaginary part of the
refractive index, which increases towards shorter wavelengths."

Reply to 2.: We changed the spelling to Chýlek and corrected the typo in the reference of
Moosmüller et al. (2018) and also added the suggested reference of Mossmüller et al. (2011) before
eq. (4).